# Development of a Quadruplex RT-qPCR for the Detection of Porcine Rotaviruses and the Phylogenetic Analysis of Porcine RVH in China

**DOI:** 10.3390/pathogens12091091

**Published:** 2023-08-28

**Authors:** Kaichuang Shi, Hongjin Zhou, Shuping Feng, Junxian He, Biao Li, Feng Long, Yuwen Shi, Yanwen Yin, Zongqiang Li

**Affiliations:** 1College of Animal Science and Technology, Guangxi University, Nanning 530005, China; zhouw1714@163.com (H.Z.); ces5734@yeah.net (J.H.); 17320146096@163.com (B.L.); shiyuwen2@126.com (Y.S.); 2Guangxi Center for Animal Disease Control and Prevention, Nanning 530001, China; fsp166@163.com (S.F.); longfeng1136@163.com (F.L.); yanwen0349@126.com (Y.Y.)

**Keywords:** porcine rotavirus, rotavirus A specie (RVA), rotavirus B specie (RVB), rotavirus C specie (RVC), rotavirus H specie (RVH), multiplex real-time quantitative RT-PCR (multiplex RT-qPCR)

## Abstract

Rotavirus A species (RVA), RVB, RVC, and RVH are four species of rotaviruses (RVs) that are prevalent in pig herds, and co-infections occur frequently. In this study, a quadruplex real-time quantitative RT-PCR (RT-qPCR) for the simultaneous detection of four porcine RVs was developed by designing specific primers and probes based on the VP6 gene of RVA, RVB, RVC, and RVH, respectively. The method showed high specificity and could only detect RVA, RVB, RVC, and RVH, without cross-reaction with other porcine viruses; showed excellent sensitivity, with a limit of detection (LOD) of 1.5 copies/µL for each virus; showed good repeatability, with intra-assay coefficients of variation (CVs) of 0.15–1.14% and inter-assay CVs of 0.07–0.96%. A total of 1447 clinical fecal samples from Guangxi province in China were tested using the developed quadruplex RT-qPCR. The results showed that RVA (42.71%, 618/1447), RVB (26.95%, 390/1447), RVC (42.92%, 621/1447), and RVH (13.68%, 198/1447) were simultaneously circulating in the pig herds, and the co-infection rate of different species of rotaviruses was found to be up to 44.01% (579/1447). The clinical samples were also detected using one previously reported method, and the coincidence rate of the detection results using two methods was more than 99.65%. The phylogenetic tree based on the VP6 gene sequences of RVH revealed that the porcine RVH strains from Guangxi province belonged to the genotype I5, which was closely related to Japanese and Vietnamese strains. In summary, an efficient, sensitive, and accurate method for the detection and differentiation of RVA, RVB, RVC, and RVH was developed and applied to investigate the prevalence of porcine RVs in Guangxi province, China. This study is the first to report the prevalence of porcine RVH in China.

## 1. Introduction

Rotaviruses (RVs) are non-enveloped RNA viruses belonging to the genus *Rotavirus* in the family *Reoviridae* and are the main pathogens causing gastroenteritis in various hosts, such as human beings, birds, and mammals [1]. The viral genome consists of 11 segments of double-stranded RNA encoding six structural proteins (VP1-4, VP6, and VP7) and five non-structural proteins (NSP1-5/6) [2]. Based on the antigenicity and sequence diversity of the structural protein VP6, RVs can be classified into nine species, namely rotavirus A specie (RVA) to RVD, and RVF to RVJ, of which RVA, RVB, RVC, and RVH are the four RVs that have been identified in pigs to date [1,3,4].

Porcine RVA was first identified in diarrheic piglets from Australia in 1975, followed by RVB and RVC in piglets from the United Kingdom in 1980s and the United States in 1980, respectively [5,6,7]. Compared to RVB and RVC, RVA is considered to be the most predominant rotavirus due to its high detection rate in different pig herds [4,8]. Porcine RVC has been identified in many countries around the world [9,10,11,12,13,14]. After 2009, the prevalence of RVC infection in animals increased from 10% to 25% [14]. In recent years, an increasing number of countries have reported high positivity rates of porcine RVC. So far, the highest positivity rate of porcine RVC, which reached to 76.10%, was reported in the United States [15]. Porcine RVB has been reported in North America, South America, Europe, Asia, the United States, Canada, Brazil, Italy, Russia, Switzerland, Japan, and India [16,17,18,19,20,21,22]. The incidence of porcine RVB is usually lower than that of RVA and RVC, and porcine RVB is usually coinfected with other RVs, such as RVA and/or RVC [23,24]. However, in 2020, Brazil reported 66.4% of the samples were only positive for RVB, and no other porcine RV was found, indicating that RVB acted as an important primary enteric pathogen in piglets [25]. One research indicated that porcine RVH has spread in the United States since 2002 [26], but porcine RVH was first isolated from diarrheic pigs in Japan in 2011 [27]. To date, RVH has been discovered in different countries, such as Italy, the United States, Brazil, South Africa, Spain, and Vietnam [18,28,29,30,31,32]. RVE was first discovered in pigs in the United Kingdom in 1986 [33], but it has not been found elsewhere and even in the United Kingdon since then, so it has been dropped from classifications. So far, porcine RVs, including RVA, RVB, RVC, and RVH, have been reported in many countries around the world, and have caused huge losses to the pig industry [1,4,8].

Porcine RVs are important diarrheic pathogens to pigs, and the piglets suffering from diarrhea were often co-infected with multiple species of porcine RVs [9,20,24]; thus, it is of great significance to develop a method for the simultaneous detection and differentiation of different groups of porcine RVs in the clinical practice. Multiplex real-time quantitative PCR (qPCR) is a commonly used technique for viral detection owing to its high specificity, sensitivity, accuracy, high-throughput, and ability to detect several viruses in one tube during the same reaction [34,35]. So far, the multiplex qPCR has been developed for detection of two species of the four porcine RVs [18,36,37]. However, no multiplex RT-qPCR has been reported for the simultaneous detection of RVA, RVB, RVC, and RVH until now. Therefore, to meet the urgent needs of clinical practice, a quadruplex RT-qPCR assay was developed for the simultaneous detection and differentiation of RVA, RVB, RVC, and RVH in this study.

## 2. Materials and Methods

### 2.1. Vaccine Strains and Positive Clinical Samples

The vaccine strains of porcine epidemic diarrhea virus (PEDV, CV777 strain), RVA (NX strain), transmissible gastroenteritis virus (TGEV, H strain), foot-and-mouth disease virus (FMDV, O/Mya98/XJ/2010 strain), swine influenza virus (SIV, TJ strain), classical swine fever virus (CSFV, C strain), porcine reproductive and respiratory syndrome virus (PRRSV, TJM-F92 strain), pseudorabies virus (PRV, Bartha-K61 strain), and porcine circovirus type 2 (PCV2, SX07 strain) were bought from China Animal Husbandary Industry Co., Ltd. (Beijing, China) and stored at −80 °C until used.

The positive samples of porcine deltacoronavirus (PDCoV), African swine fever virus (ASFV), RVB, RVC, and RVH, which were confirmed by qPCR/RT-qPCR and sequencing, were provided by Guangxi Center for Animal Disease Control and Prevention (CADC), China, and stored at −80 °C until used.

### 2.2. Collection of the Clinical Samples

A total of 1447 clinical fecal samples were collected from 1447 diarrheic pigs in 9 regions of Guangxi province, southern China, from January 2021 to December 2022. The samples were transported to the laboratory at less than 4 °C within 12 h after collection and stored at −80 °C until used.

### 2.3. Design of Primers and Probes

The VP6 gene sequences of RVA, RVB, RVC, and RVH were downloaded from GenBank of the National Center for Biotechnology Information (NCBI) (https://www.ncbi.nlm.nih.gov/nucleotide/, accessed on 25 September 2021) and then aligned using Clustal W of DNAstar Ver 6.0 (https://www.dnastar.com/, accessed on 25 September 2021) to obtain the conserved regions of the sequences. Four pairs of specific primers and corresponding TaqMan probes were designed using Oligo software (Version 7.60) (https://www.oligo.net/, accessed on 25 September 2021) (Table 1), and their specificity was confirmed using the BLAST search tool from the NCBI (https://blast.ncbi.nlm.nih.gov/Blast.cgi, accessed on 25 September 2021).

### 2.4. Extraction of Nucleic Acid

The clinical fecal swabs were mixed with phosphate-buffered saline (PBS, pH7.2) (*W*/*V*, 20%), vortexed for 5 min, and centrifuged at 12,000 rpm for 10 min at 4 °C. Total DNA and RNA are extracted from 200 µL of the supernatants or vaccine solutions using MiniBEST Viral RNA/DNA Extraction Kit Ver.5.0 (TaKaRa, Dalian, China) according to the manufacturer’s instructions, reverse transcribed to cDNA using PrimeScript II 1st Strand cDNA Synthesis Kit (TaKaRa, Dalian, China) per the manufacturer’s instructions, and then stored at –80 °C until used.

### 2.5. Construction of the Standard Plasmids

The construction of the standard recombinant plasmids was performed as described by Liu et al. in a previous report [38] with minor modifications. Briefly, the targeted fragments of VP6 gene were amplified via PCR from the cDNAs of RVA, RVB, RVC, and RVH, purified, cloned into the pMD18-T vector (TaKaRa, Dalian, China), and transformed into *E. coli* DH5α cells (TaKaRa, Dalian, China). The recombinant standard plasmids were confirmed via sequencing, and named p-RVA, p-RVB, p-RVC, and p-RVH, respectively. The OD values at 260 nm and 280 nm of the standard plasmid was determined using a NanoDrop spectrophotometer (Thermo Fisher, Waltham, MA, USA), and the concentration was calculated using the following formula: $plasmid\left(copies/µL\right)$= $\frac{\left(6.02\;\times \;10^{23}\right)\;\times \;\left(X\;ng/µL\;\times \;10^{-9}\right)}{plasmid\;length\left(bp\right)\;\times \;660}$.

### 2.6. Optimization of the Reaction Parameters

To determine the optimal reaction conditions, the quadruplex RT-qPCR experiments with different annealing temperatures (56–62 °C) and concentrations of each primer and probe (20 pmol/µL, 0.2 µL–0.5 µL) were performed using the mixture of four standard plasmids as a template. All amplification reaction were performed using One-Step PrimeScript™ RT-PCR Kit (TaKaRa, Dalian, China) in QuantStudio 6 qPCR system (ABI, Carlsbad, CA, USA). The following amplification process was used: 42 °C for 5 min; 95 °C for 10 s; and then 40 cycles of 95 °C for 5 s and 58 °C for 34 s. The fluorescence signals were automatically recorded at the end of each cycle. The optimal reaction conditions were determined based on the criteria of maximum ΔRn and minimum cycle threshold (Ct).

### 2.7. Generation of the Standard Curves

The mixture of four standard plasmids (at a ratio of 1:1:1:1) with concentrations ranging from 1.5 × 10^8^ to 1.5 × 10^2^ copies/µL (final reaction concentrations: 1.5 × 10^7^ to 1.5 × 10^1^ copies/µL) was used as a template for amplification to generate the standard curves.

### 2.8. Analytical Specificity Analysis

The specificity of the quadruplex RT-qPCR was assessed using the total DNA/RNA of RVA, RVB, RVC, RVH, PEDV, TGEV, FMDV, SIV, CSFV, PRRSV, PRV, PDCoV, ASFV, and PCV2 as templates. The four standard plasmids were used as positive controls, and the negative fecal sample and distilled water were used as negative controls.

### 2.9. Analytical Sensitivity Analysis

The mixture of four standard plasmids (at a ratio of 1:1:1:1) with concentrations ranging from 1.5 × 10^8^ to 1.5 × 10^−1^ copies/µL (final reaction concentrations: 1.5 × 10^7^ to 1.5 × 10^−2^ copies/µL) was used as a template for amplification to evaluate the sensitivity of the quadruplex RT-qPCR. The limit of detection (LOD) of the method was determined based on Ct values obtained from the templates with different concentrations.

### 2.10. Repeatability Analysis

The mixture of four standard plasmids with three concentrations of 1.5 × 10^8^, 1.5 × 10^6^, 1.5 × 10^4^ copies/µL (final reaction concentrations were 1.5 × 10^7^, 1.5 × 10^5^, 1.5 × 10^3^ copies/µL) were used as template to amplify in triplicate one day to assess the intra-assay CVs, and in three different times to assess the inter-assay CVs. The repeatability and reproducibility of the method was determined based on the CVs values.

### 2.11. Detection of RVs in Clinical Samples

The total nucleic acids were extracted from 1447 clinical fecal samples from Guangxi province in China and then used to test RVA, RVB, RVC, and RVH using the developed quadruplex RT-qPCR. In addition, these samples were also tested using the RT-qPCR developed by Ferrari et al. [18], and the coincidence rate of two methods was calculated to further evaluate the feasibility of the developed assay in this study.

### 2.12. Phylogenetic Analysis Based on RVH VP6 Gene

A total of 25 RVH positive samples from different pig farms were selected randomly for amplification of VP6 gene using a pair of primers (RVH-VP6-F: GTGACCCACAAGGATGGATCTCAT; RVH-VP6-R: GAACACTGGATCCCAGTGCGTGAC) described in the previous report [31]. The total RNAs were extracted from the clinical samples, reverse transcribed to cDNA, and then used as templates to amplify the VP6 gene using Premix Taq™ (TaKaRa, Dalian, China). The amplified products were purified, ligated into the pMD18-T vector (TaKaRa, Dalian, China), and then transformed into *E. coli* DH5α. After culturing at 37 °C for 22–24 h, the positive clones were extracted and sequenced (BGI, Shenzhen, China). Then, the sequences were verified using the NCBI BLAST tool and aligned with the reference sequences using Clustal W in MEGAX. Finally, the phylogenetic tree based on the VP6 sequences, which were obtained from this study or downloaded from GenBank of NCBI (https://www.ncbi.nlm.nih.gov/nucleotide/, accessed on 25 September 2021) as reference sequences (Table 2), was constructed using the neighbor-joining (NJ) method of MEGA11.0 (https://www.megasoftware.net/, accessed on 25 September 2021) with 1000 bootstrap replications.

## 3. Results

### 3.1. Construction of the Standard Plasmids

The VP6 gene fragments of RVA, RVB, RVC, and RVH were amplified by PCR, respectively, and used to construct the recombinant standard plasmids. The plasmids were confirmed by sequencing, and named p-RVA, p-RVB, p-RVC, and p-RVH, respectively. The original concentrations of the standard plasmids p-RVA, p-RVB, p-RVC, and p-RVH were determined to be 3.26 × 10^10^, 1.78 × 10^10^, 2.62 × 10^10^, and 4.51 × 10^10^ copies/µL, respectively. They were adjusted to the same concentration of 1.50 × 10^10^ copies/µL and stored at −80 °C until used.

### 3.2. Optimization of the Reaction Conditions

After optimization of the reaction conditions of different annealing temperatures, and different primer and probe concentrations using orthogonal tests, the optimal parameters of the quadruplex RT-qPCR were obtained. The optimal reaction system in a total volume of 20 µL is shown in Table 3. The one-step amplification parameters were as follows: 42 °C for 5 min, 95 °C for 10 s, and then 40 cycles of 95 °C for 5 s, 58 °C for 34 s. The fluorescent signals were collected at the end of each cycle. The sample with a Ct value ≤ 36 was considered the positive sample and with a Ct value > 36 was considered the negative sample.

### 3.3. Generation of the Standard Curves

The mixture of four standard plasmids with concentrations from 1.5 × 10^8^ to 1.5 × 10^2^ copies/µL (final reaction concentration: 1.5 × 10^7^ to 1.5 × 10^1^ copies/µL) were used as templates to generate the standard curves. The results showed that RVA (slope = −3.117, R^2^ = 0.998, Eff% = 103.5), RVB (slope = −3.174, R^2^ = 0.999, Eff% = 104.7), RVC (slope = −3.280, R^2^ = 0.999, Eff% = 105.6), and RVH (slope = −3.239, R^2^ = 1, Eff% = 100.4) had good correlation coefficients (R^2^ ≥ 0.998) and amplification efficiencies (E) (Figure 1).

### 3.4. Specificity Analysis

The total DNA/RNA of PEDV, TGEV, FMDV, SIV, CSFV, PRRSV, PRV, PDCoV, ASFV, and PCV2 were used to evaluate the specificity of the developed quadruplex RT-qPCR. The results showed that the assay could only detect RVA, RVB, RVC, and RVH (with fluorescent signals) (Figure 2), and not cross-detected the other viruses (without fluorescent signals), indicating good specificity of the assay.

### 3.5. Sensitivity Analysis

The mixture of the four standard plasmid was 10-fold serially diluted, and the concentrations of 1.5 × 10^8^ to 1.5 × 10^−1^ copies/µL (final reaction concentrations: 1.5 × 10^7^ to 1.5 × 10^−2^ copies/µL) were used as templates for sensitivity assessment of the assay. The results showed that the LOD of RVA, RVB, RVC, and RVH was 1.5 × 10^0^ copies/µL (final reaction concentration) (Figure 3).

### 3.6. Repeatability Analysis

The coefficient of variation (CV) in intra-assay and inter-assay of the four standard plasmids was calculated using the Ct values to evaluate the repeatability of the assay. The results showed that the intra-assay CV for repeatability and the inter-assay CV for reproducibility were 0.15–1.14% and 0.07–0.96%, respectively, indicating that the quadruplex RT-qPCR was stable and reproducible (Table 4).

### 3.7. The Prevalence of RVs in Clinical Samples

The 1447 clinical fecal samples from Guangxi province were tested using the quadruplex RT-qPCR. The results showed that the positivity rates of RVA, RVB, RVC, and RVH were 42.71% (618/1447), 26.95% (390/1447), 42.92% (621/1447) and 13.68% (198/1447), respectively (Table 5). The total infection rate, single infection rate, and co-infection rate of the samples were 65.10% (942/1447), 25.09% (363/1447) and 40.01% (579/1447), respectively. As for co-infection, the RVA + RVC co-infection showed the highest positivity rate of 10.09% (146/1447) of the dual infection, the RVA + RVB + RVC co-infection showed the highest positivity rate of 12.72% (184/1447) of the triple infection, while the RVA + RVB + RVC + RVH quadruple infection showed the positivity rate of 2.07% (30/1447).

The 1447 clinical samples were also tested using the multiplex RT-qPCR for detection of RVA-RVB and RVC-RVH reported by Ferrari et al. [18], and the positivity rates of RVA, RVB, RVC, and RVH were 42.36% (613/1447), 26.81% (388/1447), 43.26% (626/1447), and 13.68% (198/1447), respectively. The coincidence rates between these two methods were 99.65%, 99.86%, 99.65%, and 100% for RVA, RVB, RVC, and RVH, respectively (Table 6).

### 3.8. Phylogenetic Analysis Based on RVH VP6 Gene Sequences

The positive samples of RVH were randomly selected to use for further analysis of the viral genetic characterization. The VP6 gene fragments were amplified, sequenced, and analyzed. Finally, a total of 25 VP6 gene sequences of RVH were obtained and uploaded to NCBI GenBank (the accession numbers: OR039733-OR039733). The sequence alignment of porcine RVH VP6 gene revealed that the 25 gene sequences had 88.3–99.8% matching nucleotide identity with each other and had 82.4–94.1% matching nucleotide identity with other reference strains obtained from GenBank (Table 2). Phylogenetic analysis based on VP6 gene sequences revealed that all RVH strains obtained in this study belonged to genotype I5 (Figure 4). In addition, the RVH-GXQZ-2022-01 strain (accession number: OR039757), together with two RVH strains obtained from Chinese environmental samples (accession number: MK379308, MK379500), belonged to the same branch as Japanese and Vietnamese strains, while the other strains of this study belonged to another branch. These results suggested that the Chinese porcine RVH strains were more closely related to the RVH strains from Vietnam and Japan, than to the RVH strains from other countries, such as Spain, South Africa, Brazil, and the United States.

## 4. Discussion

To date, four different porcine RVs, including RVA, RVB, RVC, and RVH, have been discovered to infect pigs, and have been reported in many countries around the world [1,4,8]. A number of studies have shown that co-infections of the four porcine RVs often occur. Ferrari et al. reported that the co-infection rate of RVs in suckling pigs, weaned pigs, and fattening pigs in Italy was above 60% [18]. Baumann et al. reported that 71% of pigs in Switzerland were infected with multiple RVs, of which double infection of RVA + RVC (25%), and triple infection of RVA + RVB + RVC (36%) were the most common types of infection [20]. Molinari et al. reported that 86.4% samples from the United States had mixed RV infections, and RVA + RVB + RVC co-infections were as high as 24.3% [24]. Compared to the conventional PCR, the qPCR has the advantages of not needing electrophoresis, convenient operation, high sensitivity, excellent specificity, uneasy contamination, and high throughout [35]. Compared to the singleplex qPCR, the multiplex qPCR has the advantages of detection of several targets in one tube during the same reaction and is especially time-saving, labor-saving, cost-saving, and highly efficient [34]. Therefore, the multiplex qPCR has been widely used for the simultaneous detection of several pathogens in laboratories [34,35]. As for porcine RVs, since there exist co-infections of RVA, RVB, RVC, and RVH, it is necessary to establish a quadruplex qPCR to simultaneously detect and differentiate these four pathogens. So far, only the multiplex RT-qPCR for simultaneous detection of RVA + RVC, RVA + RVB, and RVC + RVH have been reported, and they could not simultaneously detect and distinguish three and more viruses of these four porcine RVs in clinical practice [18,36,37,38]. Therefore, four pairs of specific primers and probes were designed based on the conserved regions of the VP6 gene of RVA, RVB, RVC, and RVH, respectively, and a quadruplex RT-qPCR for the detection of these four porcine RVs was developed in this study. The developed assay showed good specificity, high sensitivity, and excellent repeatability. It could only amplify the targeted viruses, without cross-reaction with other swine viruses; It showed the LOD of 1.5 copies/µL (final reaction concentration) for all four viruses, with intra-assay CVs of 0.17–1.14% and inter-assay CVs of 0.07–0.96%. The assay was used to test the 1447 clinical samples and had a high coincidence rate of more than 99.65% with the reported reference multiplex RT-qPCR [18], which was also used to test these samples. The results verified the application of the developed assay in this study for the field samples.

The positivity rates of RVA, RVB, RVC, and RVH in the 1447 clinical samples were 42.71% (618/1447), 26.95% (390/1447), 42.92% (621/1447), and 13.68% (198/1447), respectively. All four porcine RVs were discovered in Guangxi province, and the positivity rate of rotavirus was 65.10% (942/1447), indicating that porcine RVs were widely prevalent in pig herds. Marthaler et al. reported that the positivity rates of RVA, RVB, and RVC were 62%, 33%, and 53% in the clinical samples from the United States, Mexico, and Canada, respectively [36]. In a recent report from Italy, the positivity rates of RVA, RVB, RVC, and RVH in 962 fecal samples were 53%, 45%, 43%, and 14%, respectively [18]. Our study demonstrated slightly lower or similar positivity rates of the four porcine RVs to those reports. In addition, this study revealed that co-infections were common in pig herds, with a co-infection rate of 40.01% (579/1447), which is consistent with the results from Switzerland and Brazil [20,24]. In this study, the co-infections with RVA + RVC (10.09%) and RVA + RVB + RVC (12.72%) were the predominant types of dual infection and triple infection, which was in agreement with the results of Baumann et al. [20]. Notably, RVH had a high positivity rate of 13.68% (190/1447) in the clinical samples. To our knowledge, this is the first report on the prevalence of porcine RVH in China, and the high positivity rates suggests that its harmfulness cannot be ignored.

Porcine RVs have been reported in China. The investigations of RVA showed that the positivity rate of RVA was 28.76% (65/226) in Shandong province during 2013–2014 [39], 4.3% (12/280) in Heilongjiang province in 2022 [40], 17.59% (70/398) in nine provinces during 2015–2017 [41], and 16.83% (100/394) in East China during 2017–2019 [42]. RVC has been reported in China and showed a positivity rate of 4.3% for RVC and 1.4% for RVA in Zhejiang province in 2013 [43], 18.7% (51/273) for RVC, and 7.0% (19/273) for RVA in Zhejiang, Shandong, Jiangsu, and Shanxi provinces during 2013–2019 [44], 12.5% (11/88) for RVA, and 11.4% (10/88) for RVC in Jiangsu province in 2022 [37]. However, RVB has not been reported in China. In addition, RVH has also not been reported in pigs in China until now, and only the sequences of two RVH strains, which were from environmental samples, were downloaded from NCBI GenBank (https://www.ncbi.nlm.nih.gov/nucleotide/, accessed on 16 May 2023). Since the RVH positive samples were first discovered in China, we randomly selected 25 positive samples to amplify the VP6 gene fragments, sequence, and analyze. Recently, the phylogenetic analysis using a complete-genome-based RVH genotyping system showed that RVH strains could be classified into genotypes I1 to I6 according to the VP6 gene sequence with a cut-off value of 87% [45]. In this study, 25 VP6 gene nucleotide sequences of porcine RVH strains were obtained. The phylogenetic tree based on the deduced amino acid sequences of RVH VP6 gene revealed that all 25 strains from Guangxi province belonged to the I5 genotype, and the two RVH strains from environmental samples also belonged to the I5 genotype, indicating that the Chinese strains share a common origin. In addition, the Chinese strains were most closely related to the Japanese and Vietnamese strains (Figure 4), suggesting that they might have a similar origin but have minor mutations.

## 5. Conclusions

A quadruplex RT-qPCR assay has been developed for the simultaneous detection and differentiation of RVA, RVB, RVC, and RVH, with good specificity, high sensitivity, and excellent repeatability. This method could be used to differentiate the co-infection of porcine RVs in clinical samples. In addition, RVA, RVB, RVC, and RVH showed high positivity rates in diarrheic clinical samples from Guangxi province in China. The phylogenetic tree based on the VP6 gene sequences of porcine RVH was first analyzed in China and revealed that all the Chinese strains belonged to the I5 genotype.

## Figures and Tables

**Figure 1 pathogens-12-01091-f001:**
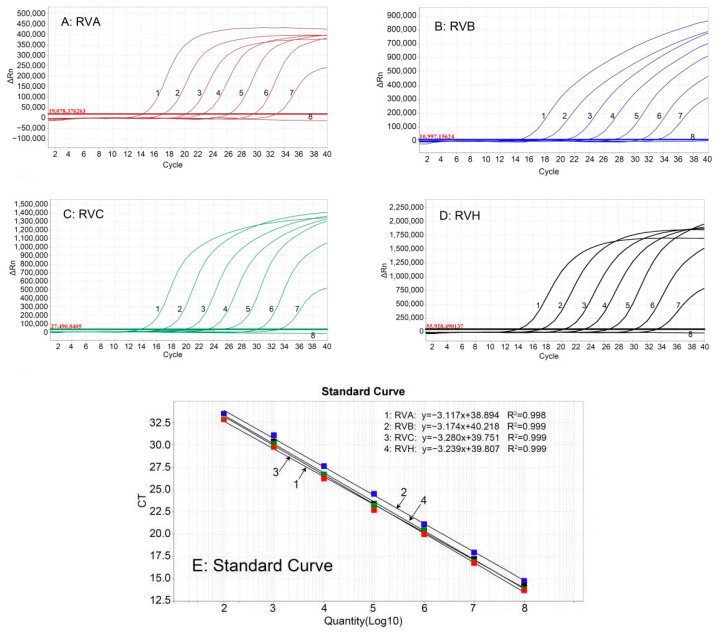
The amplification curves (**A**–**D**) and standard curves (**E**) of the quadruplex RT-qPCR. The figures show the amplification curves of the standard plasmids p-RVA (**A**), p-RVB (**B**), p-RVC (**C**), and p-RVH (**D**) with different concentrations. 1–7: The final reaction concentrations of the standard plasmids ranged from 1.5 × 10^7^ to 1.5 × 10^1^ copies/µL; 8: Negative control. The standard curves (**E**) show excellent correlation (R^2^ ≥ 0.998) between the logarithmic values of the plasmid concentrations and the Ct values.

**Figure 2 pathogens-12-01091-f002:**
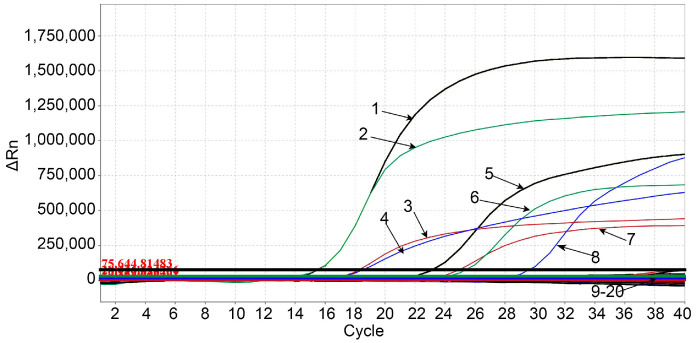
Specificity analysis of the quadruplex RT-qPCR. 1: p-RVH; 2: p-RVC; 3: p-RVA; 4: p-RVB; 5: RVH; 6: RVC; 7: RVA; 8: RVB; 9–18: PEDV, TGEV, FMDV, SIV, CSFV, PRRSV, PRV, PDCoV, ASFV, and PCV2; 19: Negative control using fecal sample; 20: Negative control using distilled water.

**Figure 3 pathogens-12-01091-f003:**
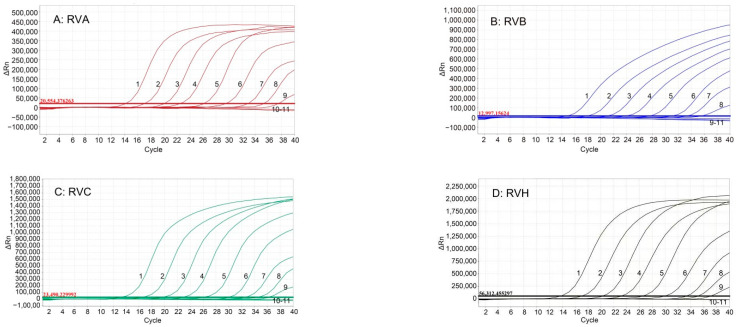
Sensitivity analysis of the quadruplex RT-qPCR. The figures show the amplification curves of the standard plasmids p-RVA (**A**), p-RVB (**B**), p-RVC (**C**), and p-RVH (**D**) with different concentrations. 1–10: The final reaction concentrations of the standard plasmids ranged from 1.5 × 10^7^ to 1.5 × 10^−2^ copies/µL. 11: Negative control using fecal sample.

**Figure 4 pathogens-12-01091-f004:**
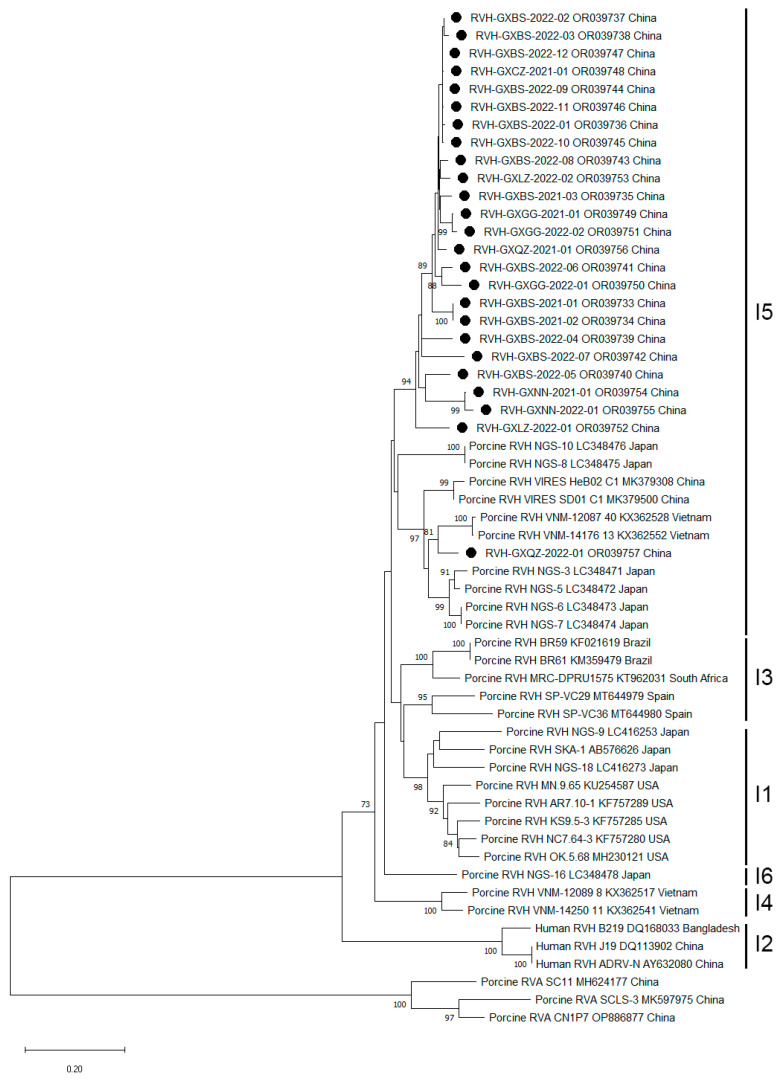
Phylogenetic tree based on the RVH VP6 amino acid sequences. The tree was constructed using the neighbor-joining method and the Kimura three-parameter model. Numbers along the tree represent the confidence value for a given internal branch based on 1000 Bootstrap replicates, and only values larger than 70 are shown. The RVH strains obtained in this study were marked with circles (●).

**Table 1 pathogens-12-01091-t001:** Primers and TaqMan probes used for detection of RVA, RVB, RVC, and RVH.

Primer/Probe	Sequence (5′→3′)	Position ^a^	Genotype	Size/bp
RVA-VP6-F	AATATGACACCAGCAGTTGCAAA	918–940	I5	107
RVA-VP6-R	ACAGATTCACAAACTGCAGATTCAA	1000–1024		
RVA-VP6-P	CY5-CAAGCACCGCCATTTATATTTCATGCTACA-BHQ3	951–980		
RVB-VP6-F	GTGTCYGCRTWTGCTGC	1181–1197	I6, I8, I10, I11, I12, I13	62
RVB-VP6-R	CCTYTCGAAGCACTYCC	1227–1243		
RVB-VP6-P	VIC-GGRAGCTGACGCCGGATCAGA-BHQ1	1203–1223		
RVC-VP6-F	GTGAAGAGAATGGTGATGTAG	1189–1209	I1, I4, I5, I6, I7, I10, I11, I12, I13	157
RVC-VP6-R	GTTCACATTTCATCCTCCTG	1324–1343		
RVC-VP6-P	FAM-TAGCATGATTCACGAATGGGTTTAG-BHQ1	1255–1279		
RVH-VP6-F	GGAAGAGCTACTGGAAAGATGG	43–64	I1, I3, I4, I5, I6	99
RVH-VP6-R	GACTCCTGAGCATGGTACTTTC	120–141		
RVH-VP6-P	Texas Red-CAGTTCAAGGCAGACCAGGAGGAA-BHQ2	77–100		

^a^ Note: The locations of the primers and probes on the VP6 gene correspond to the reference strains of RVA, RVB, RVC and RVH, whose GenBank accession numbers were FJ617209, KF882587, KC164677, and MK379512, respectively.

**Table 2 pathogens-12-01091-t002:** RVH strains used for phylogenetic analysis in this study.

No.	Strain	Accession No.	Date	Origin	Species
1	ADRV-N	AY632080	1997	China	Human
2	J19	DQ113902	1997	China	Human
3	SKA-1	AB576626	1999	Japan	Porcine
4	B219	DQ168033	2002	Bangladesh	Human
5	MRC-DPRU1575	KT962031	2007	South Africa	Porcine
6	KS9.5-3	KF757285	2008	USA	Porcine
7	OK.5.68	MH230121	2008	USA	Porcine
8	NC7.64-3	KF757280	2008	USA	Porcine
9	MN.9.65	KU254587	2008	USA	Porcine
10	AR7.10-1	KF757289	2012	USA	Porcine
11	BR59	KF021619	2012	Brazil	Porcine
12	BR61	KM359479	2012	Brazil	Porcine
13	VNM/12089_8	KX362517	2012	Vietnam	Porcine
14	VNM/14250_11	KX362541	2012	Vietnam	Porcine
15	VNM/12087_40	KX362528	2012	Vietnam	Porcine
16	VNM/14176_13	KX362552	2012	Vietnam	Porcine
17	NGS-3	LC348471	2014	Japan	Porcine
18	NGS-5	LC348472	2014	Japan	Porcine
19	NGS-6	LC348473	2014	Japan	Porcine
20	NGS-7	LC348474	2014	Japan	Porcine
21	NGS-16	LC348478	2014	Japan	Porcine
22	NGS-8	LC348475	2015	Japan	Porcine
23	NGS-9	LC416253	2015	Japan	Porcine
24	NGS-10	LC348476	2015	Japan	Porcine
25	NGS-18	LC416273	2015	Japan	Porcine
26	SP-VC29	MT644979	2017	Spain	Porcine
27	SP-VC36	MT644980	2017	Spain	Porcine
28	VIRES_HeB02_C1	MK379308	2017	China	Porcine
29	VIRES_SD01_C1	MK379500	2017	China	Porcine
30	RVH-GXBS-2021-01	OR039733	2021	China (This study)	Porcine
31	RVH-GXBS-2021-02	OR039734	2021	China (This study)	Porcine
32	RVH-GXBS-2021-03	OR039735	2021	China (This study)	Porcine
33	RVH-GXBS-2022-01	OR039736	2022	China (This study)	Porcine
34	RVH-GXBS-2022-02	OR039737	2022	China (This study)	Porcine
35	RVH-GXBS-2022-03	OR039738	2022	China (This study)	Porcine
36	RVH-GXBS-2022-04	OR039739	2022	China (This study)	Porcine
37	RVH-GXBS-2022-05	OR039740	2022	China (This study)	Porcine
38	RVH-GXBS-2022-06	OR039741	2022	China (This study)	Porcine
39	RVH-GXBS-2022-07	OR039742	2022	China (This study)	Porcine
40	RVH-GXBS-2022-08	OR039743	2022	China (This study)	Porcine
41	RVH-GXBS-2022-09	OR039744	2022	China (This study)	Porcine
42	RVH-GXBS-2022-10	OR039745	2022	China (This study)	Porcine
43	RVH-GXBS-2022-11	OR039746	2022	China (This study)	Porcine
44	RVH-GXBS-2022-12	OR039747	2022	China (This study)	Porcine
45	RVH-GXCZ-2021-01	OR039748	2021	China (This study)	Porcine
46	RVH-GXGG-2021-01	OR039749	2021	China (This study)	Porcine
47	RVH-GXGG-2022-01	OR039750	2022	China (This study)	Porcine
48	RVH-GXGG-2022-02	OR039751	2022	China (This study)	Porcine
49	RVH-GXLZ-2022-01	OR039752	2022	China (This study)	Porcine
50	RVH-GXLZ-2022-02	OR039753	2022	China (This study)	Porcine
51	RVH-GXNN-2021-01	OR039754	2021	China (This study)	Porcine
52	RVH-GXNN-2022-01	OR039755	2022	China (This study)	Porcine
53	RVH-GXQZ-2021-01	OR039756	2021	China (This study)	Porcine
54	RVH-GXQZ-2022-01	OR039757	2022	China (This study)	Porcine

**Table 3 pathogens-12-01091-t003:** The reaction system of the quadruplex RT-qPCR.

Reagent	Volume (µL)	Final Concentration (nM)
2× One Step RT-PCR Buffer III	10.0	/
Ex Taq HS (5 U/µL)	0.4	/
PrimeScript RT Enzyme Mix II	0.4	/
RVA-VP6-F	0.1	100
RVA-VP6-R	0.1	100
RVA-VP6-P	0.1	100
RVB-VP6-F	0.3	300
RVB-VP6-R	0.3	300
RVB-VP6-P	0.1	100
RVC-VP6-F	0.3	300
RVC-VP6-R	0.3	300
RVC-VP6-P	0.1	100
RVH-VP6-F	0.2	200
RVH-VP6-R	0.2	200
RVH-VP6-P	0.2	200
Nucleic acid	2.0	/
RNase Free Distilled Water	Up to 20.0	/

**Table 4 pathogens-12-01091-t004:** Repeatability analysis of the quadruplex RT-qPCR.

Plasmid	Concentration (Copies/µL)	Ct Value of Intra-Assay			Ct Value of Inter-Assay		
		x	SD	CV (%)	x	SD	CV (%)
p-RVA	1.5 × 10^7^	14.47	0.14	0.97	14.43	0.08	0.55
	1.5 × 10^5^	20.15	0.09	0.45	19.91	0.11	0.55
	1.5 × 10^3^	26.18	0.04	0.15	26.21	0.05	0.19
p-RVB	1.5 × 10^7^	14.90	0.17	1.14	14.77	0.09	0.61
	1.5 × 10^5^	21.15	0.18	0.85	21.17	0.09	0.43
	1.5 × 10^3^	27.67	0.17	0.61	27.64	0.02	0.07
p-RVC	1.5 × 10^7^	13.66	0.10	0.73	13.53	0.13	0.96
	1.5 × 10^5^	19.92	0.05	0.25	19.99	0.17	0.85
	1.5 × 10^3^	26.56	0.21	0.79	26.73	0.09	0.34
p-RVH	1.5 × 10^7^	14.02	0.11	0.78	13.93	0.06	0.43
	1.5 × 10^5^	20.36	0.06	0.29	20.26	0.06	0.30
	1.5 × 10^3^	26.67	0.09	0.34	26.68	0.18	0.67

**Table 5 pathogens-12-01091-t005:** Detection results of the clinical samples in Guangxi province in China.

Region	Number	Number of Positive Sample																
		RVA	RVB	RVC	RVH	A + B ^a^	A + C	A + H	B + C	B + H	C + H	A + B + C	A + B + H	A + C + H	B + C + H	A + B + C + H	Single Infection	Co-Infection
Chongzuo	142	45	9	46	14	0	16	0	2	1	6	3	0	1	2	1	42 (29.58%)	32 (22.54%)
Baise	594	336	183	266	80	22	54	25	11	2	6	128	2	30	3	2	128 (21.55%)	285 (47.98%)
Guigang	205	66	47	84	30	2	32	0	2	3	4	14	0	1	2	16	26 (12.68%)	76 (37.07%)
Nanning	254	91	67	145	12	0	39	3	33	2	2	26	0	2	0	2	65 (25.59)	109 (42.91%)
Hechi	12	6	3	2	0	1	0	0	0	0	0	2	0	0	0	0	3 (25.00%)	3 (25.00%)
Qinzhou	67	13	35	22	8	0	1	0	6	4	0	8	0	3	0	1	19 (28.36%)	23 (34.33%)
Beihai	37	0	0	0	13	0	0	0	0	0	0	0	0	0	0	0	13 (35.14%)	0 (0%)
Liuzhou	111	59	36	42	34	9	3	3	1	0	6	3	2	8	0	8	56 (50.45%)	43 (38.74%)
Yulin	25	2	10	14	7	1	1	0	0	0	0	0	0	0	6	0	11 (44.00%)	8 (32.00%)
Total	1447	618 (42.71%)	390 (26.95%)	621 (42.92%)	198 (13.68%)	35 (2.42%)	146 (10.09%)	31 (2.14%)	55 (3.80%)	12 (0.83%)	24 (1.66%)	184 (12.72%)	4 (0.28%)	45 (3.11%)	13 (0.90%)	30 (2.07%)	363 (25.09%)	579 (40.01%)

^a^ Note: A + B stands for co-infection of RVA and RVB; A + C stands for co-infection of RVA and RVC; A + H stands for co-infection of RVA and RVH; B + C stands for co-infection of RVB and RVC; B + H stands for co-infection of RVB and RVH; C + H stands for co-infection of RVC and RVH; A + B + C stands for co-infection of RVA, RVB, and RVC; A + B + H stands for co-infection of RVA, RVB, and RVH; A + C + H stands for co-infection of RVA, RVC, and RVH; B + C + H stands for co-infection of RVB, RVC, and RVH; A + B + C + H stands for co-infection of RVA, RVB, RVC, and RVH.

**Table 6 pathogens-12-01091-t006:** Agreements of the detection results using the developed and the reported reference multiplex RT-qPCR.

Method	Number of Positive Samples			
	RVA (%)	RVB (%)	RVC (%)	RVH (%)
The Developed Multiplex RT-qPCR	618/1447 (42.71%)	390/1447 (26.95%)	621/1447 (42.92%)	198/1447 (13.68%)
The Reported Reference Method	613/1447 (42.36%)	388/1447 (26.81%)	626/1447 (43.26%)	198/1447 (13.68%)
Agreements	99.65%	99.86%	99.65%	100.00%

## Data Availability

Not applicable.

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
