# Peer review of "Development of a Quadruplex RT-qPCR for the Detection of Porcine Rotaviruses and the Phylogenetic Analysis of Porcine RVH in China"

_pathogens, 2023, doi:10.3390/pathogens12091091_

Round 1

Reviewer 1 Report

The study describe the development of quadruplex RT-qPCR assay for detection of 4 rotavirus species (A, B, C and H). This will be of great importance in surveillance of rotaviruses in swine industry leading to instituting effective control and prevention strategies for RV infections. The study detected RVH in China swine population for the first time, analyzed the VP6 gene of the detected RVH strains and compared the RVH sequences already available in public database. The MS is well written and the assay well described. My only main concern relates to the phylogenetic tree (Fig. 4), where the bootstrap values for branches separating I5, I6, I4, I3 and I1 is too low meaning the method used to construct the phylogenetic tree is not the best method. Hence, I advise that the authors should evaluate which method will give the best tree. Additionally, the authors should present the phylogenetic tree for the deduced amino acid sequences for these strains.

Minor comments

1. The acronym for real time (quantitative) RT-PCR should be RT-qPCR and not qRT-PCR. The authors should correct this throughout the MS. (NB: RT=reverse transcription)

2. L39 - Currently, RV is classified into species not groups and so far there are 9 species (A-D, F-J). RVE has been dropped from classification because since it detection 1986 in UK it has not been found elsewhere and even in UK. -- authors should be aware of this, and change all "groups" to "species" in the MS.

3. L111 - replace "reverse transferred to ...." to "reverse transcribed to ...."

4. L156 - replace the section heading with "Detection of RVs in clinical samples"

5. L167 - replace " reverse transferred to cDNA ..." to "reverse transcribed to cDNA ..."

6. L237 - replace section head with " prevalence of RVs in clinical samples"

7. L278 - Fig. 4 please refer to my main comments above. Also provide full details of the method used in the caption. The authors should use outgroup to root the tree.

The English is sufficient but require a little improvement especially section titles

Author Response

The Cover Letter (#2)

August 21, 2023

We have revised our manuscript carefully according to the editor’s and reviewers’s suggestions. The detail is as follows.

Reviewer 1

Comments and Suggestions for Authors

The study describes the development of quadruplex RT-qPCR assay for detection of 4 rotavirus species (A, B, C and H). This will be of great importance in surveillance of rotaviruses in swine industry leading to instituting effective control and prevention strategies for RV infections. The study detected RVH in China swine population for the first time, analyzed the VP6 gene of the detected RVH strains and compared the RVH sequences already available in public database. The MS is well written and the assay well described. My only main concern relates to the phylogenetic tree (Fig. 4), where the bootstrap values for branches separating I5, I6, I4, I3 and I1 is too low meaning the method used to construct the phylogenetic tree is not the best method. Hence, I advise that the authors should evaluate which method will give the best tree. Additionally, the authors should present the phylogenetic tree for the deduced amino acid sequences for these strains.

Response: The phylogenetic tree is re-constructed using the neighbor-joining method, basing on the deduced amino acid sequences. Please see the Figure 4 in the revised manuscript.

Minor comments

  1. The acronym for real time (quantitative) RT-PCR should be RT-qPCR and not qRT-PCR. The authors should correct this throughout the MS. (NB: RT=reverse transcription).

Response: “qRT-PCR” has been changed to “RT-qPCR” through the revised manuscript.

  1. L39 - Currently, RV is classified into species not groups and so far there are 9 species (A-D, F-J). RVE has been dropped from classification because since it detection 1986 in UK it has not been found elsewhere and even in UK. -- authors should be aware of this, and change all "groups" to "species" in the MS.

Response: Done. Please see Lines 11, 22, 31, 32, 41, 42, and 74 in the revised manuscript.

  1. L111 - replace "reverse transferred to ...." to "reverse transcribed to ...."

Response: Done. Please see Line 116 in the revised manuscript.

  1. L156 - replace the section heading with "Detection of RVs in clinical samples"

Response: Done. Please see Line 161 in the revised manuscript.

  1. L167 - replace " reverse transferred to cDNA ..." to "reverse transcribed to cDNA ..."

Response: Done. Please see Line 172 in the revised manuscript.

  1. L237 - replace section head with " prevalence of RVs in clinical samples"

Response: Done. Please see Line 243 in the revised manuscript.

  1. L278 - Fig. 4 please refer to my main comments above. Also provide full details of the method used in the caption. The authors should use outgroup to root the tree.

Response: The phylogenetic tree is re-constructed using the neighbor-joining method, basing on the deduced amino acid sequences. Porcine RVA strains have been used as the outgroup to root the tree. Please see the Figure 4 in the revised manuscript.

Reviewer 2 Report

In this manuscript, the authors successfully established a quadruplex qRT-PCR assay for dicriminable detection of RVA, RVB, RVC, and RVH. The detection of clinical samples showed the co-infection of RVs was frequent among field samples. The phylogenetic tree based on the VP6 gene sequences of porcine RVH was also constructed and revealed that all the Chinese strains belonged to I5 genotype. Here are also other issues that need to be addressed

1. In Figure 3, the reviewer suggests to include the results of all four figures also in the same figure as figure 2.

2. In Figure 4, the strains LC348475 and LC348476 were clustered into I1 with other strains. However, these two strains were significantly far related with I1. The mistake should be collected.

3. In Figure 4, most of the sequenced samples were from Baise, and much few were from other cities, no samples from Yulin. The selection of samples should consider their distribution but not randomly. Importantly, a strain far related with others was identified in Qinzhou. The reviewer suggested to sequence more strains from other cities in Guangxi.

4. Several writing mistakes were present in the manuscript and should be corrected.

A. Line 134, Ct and line 194, Ct ?

B. Line 176, 1000 bootstrap should be 1, 000

OK

Author Response

The Cover Letter (#2)

August 21, 2023

We have revised our manuscript carefully according to the editor’s and reviewers’ suggestions. The detail is as follows.

Reviewer 2

Comments and Suggestions for Authors

In this manuscript, the authors successfully established a quadruplex qRT-PCR assay for discriminable detection of RVA, RVB, RVC, and RVH. The detection of clinical samples showed the co-infection of RVs was frequent among field samples. The phylogenetic tree based on the VP6 gene sequences of porcine RVH was also constructed and revealed that all the Chinese strains belonged to I5 genotype. Here are also other issues that need to be addressed

  1. In Figure 3, the reviewer suggests to include the results of all four figures also in the same figure as figure 2.

Response: We believe that presenting the results of sensitivity analysis separately using A, B, C, and D graphs would be more accurate and intuitive, so we keep the Figure 3 unchanged.

  1. In Figure 4, the strains LC348475 and LC348476 were clustered into I1 with other strains. However, these two strains were significantly far related with I1. The mistake should be collected.

Response: The phylogenetic tree is re-constructed based on the deduced amino acid sequence of VP6 gene, and the strains LC348475 and LC348476 belong to I5 genotype. Please see the Figure 4 in the revised manuscript.

  1. In Figure 4, most of the sequenced samples were from Baise, and much few were from other cities, no samples from Yulin. The selection of samples should consider their distribution but not randomly. Importantly, a strain far related with others was identified in Qinzhou. The reviewer suggested to sequence more strains from other cities in Guangxi.

Response: We selected the positive samples from Yulin to amplify, sequence, and analyze the VP6 gene. The results showed that the strains from Yulin also belonged to I5 genotype. Since we have not uploaded those sequences to NCBI GenBank yet, and have not obtained the accession numbers. Therefore, the phylogenetic tree in this manuscript did not include the sequences from Yulin.

A strain of RVH from Qinzhou is far related with other strains, indicating that there exists genetic variation. This need to be further studied.

  1. Several writing mistakes were present in the manuscript and should be corrected.
  2. Line 134, Cand line 194, Ct?

Response: Ct was used. Please see Lines 139, 199, and 200 in the revised manuscript.

  1. Line 176, 1000 bootstrap should be 1, 000.

Response: Done. Please see Line 182 in the revised manuscript.

Reviewer 3 Report

In the present manuscript, the authors have optimized a quadruplex quantitative Reverse Transcription Polymerase Chain Reaction (qRT-PCR) method to identify the co-infection from Rotaviruses. This includes the detection of various strains such as Group A rotavirus (RVA), RVB, RVC, and RVH. The refined assay also proves valuable in distinguishing co-infections and differentiations. The manuscript is effectively composed and adequately supported with essential control experiments.

I do not have any concern with the assay design or obtained conclusion from it

Require Minor editing.

Author Response

The Cover Letter (#2)

August 21, 2023

We have revised our manuscript carefully according to the editor’s and reviewers’ suggestions. The detail is as follows.

Reviewer 3

Comments and Suggestions for Authors

In the present manuscript, the authors have optimized a quadruplex quantitative Reverse Transcription Polymerase Chain Reaction (qRT-PCR) method to identify the co-infection from Rotaviruses. This includes the detection of various strains such as Group A rotavirus (RVA), RVB, RVC, and RVH. The refined assay also proves valuable in distinguishing co-infections and differentiations. The manuscript is effectively composed and adequately supported with essential control experiments.

I do not have any concern with the assay design or obtained conclusion from it

Response: Thanks very much.

Reviewer 4 Report

This manuscript describes development of multiplex RT-qPCR to simultaneously detect four species of porcine RVs distributed in many countries. The developed quadruplex RT-qPCR has similar and/or high sensitivity, specificity, and reproducibility compared to single-, double-, and triple-plex RT-qPCR. In addition, the authors demonstrate single and co-infections of porcine RVs from clinical samples collected in Guangxi province using this methodology. Therefore, it is suitable for publication in this journal. However, this manuscript needs minor revisions before publication.

Minor comments:

Group A rotavirus→ Rotavirus A specie

2.3. Design of primers and probes

Currently, porcine RVA, RVB, RVC and RVH have multiple genotypes, respectively.

Did the authors confirm the designed primers and probe can detect RVs with all genotypes in silico? If not check all genotypes, the authors should show genotypes can detect using this methodology in each RV specie.

Table 1

The authors should show positions of primers and probe in the used reference strain in addition to size, respectively.

Table 4

The authors should improve Table 4 by adding horizontal lines to understand better it.

Figure 1-3

The authors should change yellow into other color to understand better the graph.  

Author Response

The Cover Letter (#2)

August 21, 2023

We have revised our manuscript carefully according to the editor’s and reviewers’s suggestions. The detail is as follows.

Reviewer 4

Comments and Suggestions for Authors

This manuscript describes development of multiplex RT-qPCR to simultaneously detect four species of porcine RVs distributed in many countries. The developed quadruplex RT-qPCR has similar and/or high sensitivity, specificity, and reproducibility compared to single-, double-, and triple-plex RT-qPCR. In addition, the authors demonstrate single and co-infections of porcine RVs from clinical samples collected in Guangxi province using this methodology. Therefore, it is suitable for publication in this journal. However, this manuscript needs minor revisions before publication.

Minor comments:

  1. Group A rotavirus→ Rotavirus A specie

Response: Done. Please see Lines 11, and 31 in the revised manuscript.

  1. 2.3. Design of primers and probes

Currently, porcine RVA, RVB, RVC and RVH have multiple genotypes, respectively.

Did the authors confirm the designed primers and probe can detect RVs with all genotypes in silico? If not check all genotypes, the authors should show genotypes can detect using this methodology in each RV specie.

 Response: The main genotypes of RVs detected by the designed primers and probes were added to Table 1. Please see Table 1 in the revised manuscript.

  1. Table 1

The authors should show positions of primers and probe in the used reference strain in addition to size, respectively.

 Response: The positions of primers and probe in the used reference strain have been added to Table 1. Please see Table 1 in the revised manuscript.

  1. Table 4

The authors should improve Table 4 by adding horizontal lines to understand better it.

 Response: Done. Please see Table 4 in the revised manuscript.

  1. Figure 1-3

The authors should change yellow into other color to understand better the graph.  

Response: Done. We have changed yellow to black. Please see Figure 1 to Figure 3 in the revised manuscript.